# Fast Abductive Learning by Similarity-based Consistency Optimization

**Yu-Xuan Huang**[1], **Wang-Zhou Dai**[2], **Le-Wen Cai**[1], **Stephen Muggleton**[2], **Yuan Jiang**[1]

[1]National Key Laboratory for Novel Software Technology
Nanjing University, Nanjing 210023, China
`{huangyx, cailw, jiangy}@lamda.nju.edu.cn`
[2]Department of Computing, Imperial College London, London SW7 2AZ, UK
`{w.dai, s.muggleton}@imperial.ac.uk`

## Abstract

To utilize the raw inputs and symbolic knowledge simultaneously, some recent neuro-symbolic learning methods use abduction, i.e., abductive reasoning, to integrate sub-symbolic perception and logical inference. While the perception model, e.g., a neural network, outputs some facts that are inconsistent with the symbolic background knowledge base, abduction can help revise the incorrect perceived facts by minimizing the inconsistency between them and the background knowledge. However, to enable effective abduction, previous approaches need an initialized perception model that discriminates the input raw instances. This limits the application of these methods, as the discrimination ability is usually acquired from a thorough pre-training when the raw inputs are difficult to classify. In this paper, we propose a novel abduction strategy, which leverages the similarity between samples, rather than the output information by the perceptual neural network, to guide the search in abduction. Based on this principle, we further present ABductive Learning with Similarity (ABLSim) and apply it to some difficult neuro-symbolic learning tasks. Experiments show that the efficiency of ABLSim is significantly higher than the state-of-the-art neuro-symbolic methods, allowing it to achieve better performance with less labeled data and weaker domain knowledge.

## 1 Introduction

To address the limitations of current machine learning methods, the next generation of Artificial Intelligence calls for the integration of data-driven machine learning and knowledge-driven reasoning such as logic inference [1]. Neuro-Symbolic Learning [8, 22] and Statistical Relational AI [23] are representative works in this direction. However, most of these approaches try to approximate logical calculus with differentiable functions using distributed representations in a neural network, and train the model in an end-to-end manner, which usually demand a large number of labeled data.

Abductive Learning (ABL) [5, 29] is a novel framework attaching machine learning models to a first-order logical reasoning model while preserving the full expressive power of each side: The machine learning model learns to convert raw input data (e.g., images, text) into symbolic representations, and the logical model tries to reason about them. Because ABL allows full-featured logical reasoning, it can directly consult symbolic background knowledge bases and reduce the requirement for massive labeled data. The logical reasoning model adopts *abductive reasoning* [20], or *abduction*, to search for the labels of the unlabeled instances, which are used for updating the machine learning model. Because abductive reasoning is non-deterministic, for each unlabeled instance there could be multiple *abduced labels*. To choose the best labels, one needs to *minimize the inconsistency* between the abduced labels and the symbolic background knowledge.

35th Conference on Neural Information Processing Systems (NeurIPS 2021).

Hence, a well-designed consistency measure improves the quality of the abduced labels and leads to a high-performance model. For example, some approaches take the consistency score as the size of the largest subset of unlabeled examples with abduced labels that are consistent with knowledge base, leading to a subset-selection problem that is difficult to solve [5, 29]; some other works measure the confidence of the predicted labels by the perceptual machine learning model, which could be unreliable when the model is under-trained [19, 4, 2].

According to its definition, *abduction* refers to the process of inferring specific facts that give the best explanation to observations based on background knowledge [20]. Hence, not only the symbolized information—the labels predicted by perception model and the symbolic background knowledge base, but also the observed raw representation of the inputs contribute to the goodness of abduced labels. For example, when babies start learning an unknown language, although they could not make sense of the acoustic syllables in a sentence, they can amazingly learn from a handful of examples and understand a few words by distinguishing different sound patterns (raw representation) as well as identifying frequently occurred syllable combinations (symbolic relations) [9].

Inspired by this phenomenon, we develop a similarity-based consistency measure for abduction, which takes the idea that samples in the same category are similar in feature space while samples of different classes are dissimilar. It can be regarded as the clustering initialization for the perception model [29], and then the metric for clustering in embedding space is improved when the model gets updated by the abduced labels during training. Applying this principle, we propose ABductive Learning with Similarity (ABLSim), which adopts beam search to solve the optimization problem of finding the best abduced labels. We verify the effectiveness of ABLSim on four neuro-symbolic tasks. Compared with other methods, ABLSim can abduce higher quality labels for unlabeled data and accelerate the model training, bringing a significantly better performance. Moreover, even when we increase the difficulty of abduction by removing some rules from the knowledge base, ABLSim still achieves comparable results as the other abduction-based neuro-symbolic learning methods with a full knowledge base.

## 2  Related Work

Neuro-symbolic (NeSy) learning  [8, 22] proposes to enhance machine learning with symbolic reasoning. It tries to learn the ability for both perception from the environment and reasoning from what has been perceived.  Most of the approaches model this pipeline with an end-to-end deep neural network, in which a symbolic domain knowledge base is used for building the structure of neural networks [8, 25, 11, 10, 24, 27]. However, most of these methods replace the symbolic representation with distributed representation and approximate logic inference with fuzzy operations. As a consequence, they usually require a large amount of labeled training data and are difficult to extrapolate.

Probabilistic Logic Program (PLP) [6] and Statistical Relational Learning (SRL) [15, 23] supply symbolic models with a probabilistic semantic and preserve the logical formulation. PLP extends first-order logic to accommodate probabilistic groundings to include probabilistic inference; SRL tries to leverage domain knowledge to construct a probabilistic graphical model structure for statistical inference. They usually require direct semantic level input and are difficult to be applied to raw input data such as images. A typical exception in this area is DeepProbLog [21], which unifies probabilistic logical inference with neural network training by gradient descent. However, due to the exponential complexity of the probabilistic distribution on the Herbrand base, the probabilistic inference in these methods could be inefficient for complicated tasks.

Recently, some approaches leverage abduction to train machine learning and perform logical reasoning simultaneously, e.g., the Abductive Learning (ABL) [5, 29] framework. ABL uses machine learning models to predict pseudo-labels for the unlabeled data, and then uses first-order logical abduction to revise them and update the machine learning model. The abduction searches for a set of revised pseudo-labels to minimize the inconsistency between data and knowledge base. However, when the model is under-trained, the revision becomes difficult because the predicted pseudo-labels are mostly incorrect, and the abduction could be easily trapped in local optima [4, 2]. The Neural-Grammar-Symbolic model (NGS) [19] shares a similar idea of abduction. It takes context-free grammar as the knowledge base and uses Markov Chain Monte Carlo (MCMC) sampling to revise the pseudo-labels according to their posterior, which suffers from the same problem when the neural

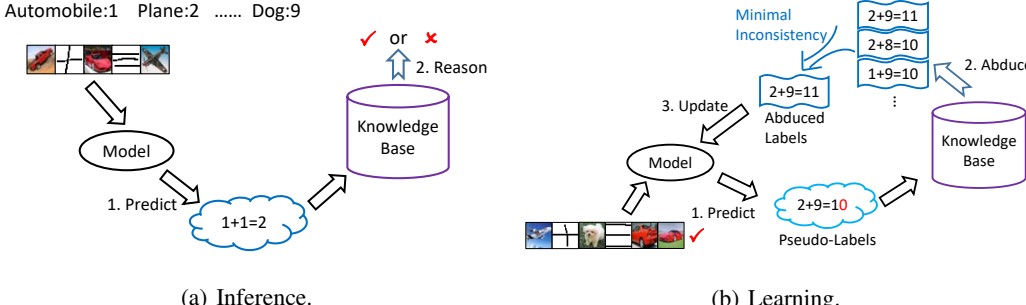

(a) Inference.  (b) Learning.

Figure 1: An example of the inference and learning process of ABL. In the inference process, the input data is a decimal equation represented by several CIFAR-10 images, e.g., the images "plane" represent digit 1, "automobile" for digit 2, "dog" for digit 9. The final output is the reasoning result on both symbols predicted by the perception model and the knowledge base. In the learning stage, the reasoning model abduces several consistent revised pseudo-labels and searches for the best one by minimizing the inconsistency, which are then used to update the perception model.

network is under-trained. Different from these approaches, ABLSim considers the similarity between input examples in the feature space rather than the output space.

# 3 Abductive Learning

## 3.1 Inference

The Abductive Learning (ABL) framework contains a perception model and a reasoning model. The perception model is used for mapping the raw input data $x \in \mathcal{X}$ into discrete symbols $z \subseteq \mathcal{Z}$, where $\mathcal{X}$ is the input space and $\mathcal{Z}$ is the output space of the perception model; $x$ and $z$ are the input *example* and extracted symbols, respectively, where $x = \{x_1, x_2, \ldots\}$ and $z = \{z_1, z_2, \ldots\}$ are the bags of meaningful *instances* detected from $x$ and the corresponding symbolic labels. For the example in Figure 1, an input image of the equation is $x$; the segmented instances of CIFAR-10 images and operation symbols are $x_i$; the sequence of predicted equation string is $z$, with each of its characters as $z_i$; whether the equation satisfies $KB$ is denoted as $y = True$ or $y = False$.

The reasoning model contains knowledge base $KB$ of first-order logic rules, which receives the symbols $z$ and reasons about the final output $y \in \mathcal{Y}$. Note that $\mathcal{Z}$ and $\mathcal{Y}$ are subsets of the Herbrand base of $KB$, i.e., elements in $\mathcal{Z} \cup \mathcal{Y}$ are ground facts of the predicates defined in $KB$. For example, given the symbolic rules of decimal arithmetic, the reasoning model would output a ground fact $y = True$ when the input fact is $z = $"1+1=2", and a ground fact $y = False$ when input is $z = $"1+9=10". Figure 1(a) gives an example of the inference process in the *CIFAR-10 Decimal Equation Decipherment* task (cf. Section 6.3).

## 3.2 Abductive Reasoning

*Abduction* (i.e. *abductive reasoning*) [14, 20] is a basic form of logical inference that seeks the best explanation for observations based on implication. It is a non-deterministic process that may have multiple answers. For example, when observing an arithmetic puzzle "$dog + cat = 2$", based on our background knowledge in mathematics, we could explain it with three abduced facts: "$dog = cat = 1$", "$dog = 2 \wedge cat = 0$" or "$dog = 0 \wedge cat = 2$".

## 3.3 Learning

Formally, given unlabeled data $\boldsymbol{X} = \{\boldsymbol{x}^{\langle 1 \rangle}, \boldsymbol{x}^{\langle 2 \rangle}, \ldots\}$, the knowledge base $KB$ and the final desired output $Y = \{y^{\langle 1 \rangle}, y^{\langle 2 \rangle}, \ldots\}$, the target is to learn a perception model $f : \mathcal{X} \mapsto \mathcal{Z}$ that accurately predicts the labels of the input instances that together with $KB$ entails $y$. Since we do not have any supervision on $z$, we call it *pseudo-label* just like in *weakly-supervised learning* [28].

The ABL framework involves three steps—predict pseudo-labels, abduce revised labels and update the model. First, the perception model $f$ is used to obtain the symbolic predictions $z = f(\boldsymbol{x})$ as pseudo-labels. Then ABL revises the pseudo-labels $z$ to $\bar{z}$ by abductive reasoning. Usually, there are several candidates $\bar{z}$ that are consistent with $KB$. The reasoning model abduces the most likely correct pseudo-labels $\bar{z}$ under the principle of *minimal inconsistency* between the data and knowledge base. Finally, ABL treats the $\bar{z}$ as ground-truth labels to update the perception model, and the above routine repeats iteratively.

Figure 1(b) gives an example, where the ground-truth labels of the input are "2+9=11" and $y = True$. The perception model $f$ predicts the wrong pseudo-labels $z =$ "2+9=10". After minimizing the inconsistency in abduction, $\bar{z} =$ "2+9=11" is selected as the final revised label to update $f$.

### 3.4 Consistency Measure

Consistency measure for abduction plays an important role in ABL. It is used to guide the search of the abduced labels $\bar{z}$ from the set of consistent candidate labels $\mathbb{A} = \{\bar{z} \mid KB \cup \bar{z} \models y\}$, where $\models$ means logical entailment. Because $\bar{z}$ will be treated as ground-truth to update the perception model, whether and how the model improves mainly depends on the design of this consistency measure. Current consistency measures mainly depend on the model's output labels or confidence, which could perform poorly when $f$ is inaccurate or not properly initialized [4, 2]. We verify this in our experiments in Section 6.

## 4 The Similarity-based Consistency Measure

We propose a similarity-based consistency measure for abduction, which takes the idea that samples in the same category are similar in feature space while samples of different classes are dissimilar. It is motivated by how humans perform abductive reasoning. Take the example in Figure 1(b). If we misclassified the last image to digit 0 and the prediction becomes "2+9=10", which is inconsistent with the knowledge base. Revisions "1+9=10", "2+8=10" and "2+9=11" are all consistent revisions. By looking at the images directly, we could observe that the last two images are similar, while the other pairwise combinations of these images look quite different. Therefore, the last two images probably share the same label. Finally, we conclude that "2+9=11" are the most likely correct labels, which could be regarded as the best abduced labels.

Given the final output $y$ and knowledge base $KB$, let $\mathbb{A}$ be the set of all consistent candidates by abductive reasoning, i.e., $\mathbb{A} = \{\bar{z} \mid KB \cup \bar{z} \models y\}$. The consistency optimization problem is equivalent to selecting the best revised labels $\bar{z}$ in $\mathbb{A}$, which can be formalized as follows:

$$\max_{\bar{z} \in \mathbb{A}} \quad \text{SimilarityScore}(\boldsymbol{x}, \bar{z}), \tag{1}$$

where the SimilarityScore measures the consistency of the revised labels $\bar{z}$. We define the consistency as the average difference between each sample's inter-class distance and intra-class distance:

$$\text{SimilarityScore}(\boldsymbol{x}, \bar{z}) = \frac{1}{\mid \boldsymbol{x} \mid} \sum_{x_i \in \boldsymbol{x}} \left(\text{InterclassDis}(x_i, \bar{z}) - \text{IntraclassDis}(x_i, \bar{z})\right). \tag{2}$$

The inter-class distance $\text{InterclassDis}(x_i, \bar{z})$ of sample $x_i$ stands for the average distance of other instances $x_j$ that have different revised labels as $x_i$; while the intra-class distance $\text{IntraclassDis}(x_i, \bar{z})$ defines the average distance of other instances $x_j$ that have the same revised labels as $x_i$, i.e.,

$$\text{InterclassDis}(x_i, \bar{z}) = \frac{1}{\mid \mathbb{D}_{i,\bar{z}} \mid} \sum_{x_j \in \mathbb{D}_{i,\bar{z}}} \text{Dis}(x_i, x_j), \tag{3}$$

$$\text{IntraclassDis}(x_i, \bar{z}) = \frac{1}{\mid \mathbb{S}_{i,\bar{z}} \mid} \sum_{x_j \in \mathbb{S}_{i,\bar{z}}} \text{Dis}(x_i, x_j), \tag{4}$$

where $\mathbb{D}_{i,\bar{z}} = \{x_j \mid \bar{z}_j \neq \bar{z}_i, j \neq i\}$ is the set of instances whose labels are *different* from $x_i$'s, and $\mathbb{S}_{i,\bar{z}} = \{x_j \mid \bar{z}_j = \bar{z}_i, j \neq i\}$ is the set of the other instances that share the *same* revised labels as $x_i$. Note that $\mathbb{D}_{i,\bar{z}} \cup \mathbb{S}_{i,\bar{z}} \cup x_i = \boldsymbol{x}$. $\text{Dis}(x_i, x_j)$ is the distance between $x_i$ and $x_j$, which will be explained later. If $\mid \mathbb{D}_{i,\bar{z}} \mid = 0$ or $\mid \mathbb{S}_{i,\bar{z}} \mid = 0$, where the denominator becomes 0, we use the

**Algorithm 1** ABLSim Learning

---

**Input:** Unlabeled data $\boldsymbol{X} = (\boldsymbol{x}^{\langle 1 \rangle}, \boldsymbol{x}^{\langle 2 \rangle}, \cdots, \boldsymbol{x}^{\langle m \rangle})$; Final output $y = (y^{\langle 1 \rangle}, y^{\langle 2 \rangle}, \cdots, y^{\langle m \rangle})$; Current model $f$; Knowledge base $KB$; Beam width (beam size of beam search) $b$
**Output:** Model $f$
1: **for** $t = 1$ **to** $T$ **do**
2:     $\mathbb{A} \leftarrow []$ # the candidate labels
3:     **for** $k = 1$ **to** $m$ **do**
4:         $\boldsymbol{z}^{\langle k \rangle} \leftarrow f(\boldsymbol{x}^{\langle k \rangle})$    # generate pseudo-labels
5:         $\mathbb{A}^{\langle k \rangle} \leftarrow \text{Abduce}(KB, \boldsymbol{z}^{\langle k \rangle}, y^{\langle k \rangle})$    # abduce all consistent revised pseudo-labels
6:         $\mathbb{A} \leftarrow \mathbb{A} \times \mathbb{A}^{\langle k \rangle}$ # Cartesian product
7:         $\boldsymbol{x} \leftarrow \boldsymbol{X}[1 : k]$
8:         $\boldsymbol{score} \leftarrow []$ # the score of each candidate labels
9:         **for** $\bar{z} \in \mathbb{A}$ **do**
10:             $\boldsymbol{score}.append(\text{Score}(\boldsymbol{x}, \bar{\boldsymbol{z}}))$ # get the score of candidate labels according to Eq. (12)
11:         **end for**
12:         $\mathbb{A} \leftarrow \text{TopN}(\mathbb{A}, \boldsymbol{score}, b)$ # select '$b$' candidate labels in $\mathbb{A}$ with the highest $\boldsymbol{score}$
13:     **end for**
14:     $\bar{\boldsymbol{Z}} \leftarrow \text{TopN}(\mathbb{A}, \boldsymbol{score}, 1)$ # select the label in $\mathbb{A}$ with the highest $\boldsymbol{score}$
15:     $f \leftarrow \text{Update}(f, \boldsymbol{X}, \bar{\boldsymbol{Z}})$    # update model $f$ using data $\boldsymbol{X}$ and abduced labels $\bar{\boldsymbol{Z}}$
16: **end for**

---

average inter-class/intra-class distance of other $x_j$ as the estimated distance of $x_i$. The details on the calculation when the denominator becomes 0 are shown in the supplementary material.

Combining (1)-(4), the similarity-based consistency measure optimization problem becomes:

$$\max_{\bar{\boldsymbol{z}} \in \mathbb{A}} \quad \frac{1}{\mid \boldsymbol{x} \mid} \sum_{x_i \in \boldsymbol{x}} \left( \frac{1}{\mid \mathbb{D}_{i,\bar{\boldsymbol{z}}} \mid} \sum_{x_j \in \mathbb{D}_{i,\bar{\boldsymbol{z}}}} \text{Dis}(x_i, x_j) - \frac{1}{\mid \mathbb{S}_{i,\boldsymbol{z}} \mid} \sum_{x_j \in \mathbb{S}_{i,\bar{\boldsymbol{z}}}} \text{Dis}(x_i, x_j) \right). \quad (5)$$

The distance $\text{Dis}(x_i, x_j)$ in (3)-(5) measures the similarity between instances $x_i$ and $x_j$, where the higher the similarity, the smaller the distance. It can be represented as follows:

$$\text{Dis}(x_i, x_j) = \text{Distance}(\phi(x_i), \phi(x_j)). \quad (6)$$

where $\phi$ is the feature map function. For example, for images or text, we can use a neural network as $\phi$ to extract the embedding for samples; for tabular data, $\phi$ can be the normalization function. In practice, we can use any appropriate metrics in (6). In this work, we use the cosine distance in our implementation, i.e., $\text{Dis}(x_i, x_j) = CosineDistance(\phi(x_i), \phi(x_j))$.

Obviously, maximizing the score in (2) means maximizing the average gap between $\text{InterclassDis}(x_i, \bar{\boldsymbol{z}})$ and $\text{IntraclassDis}(x_i, \bar{\boldsymbol{z}})$. In other words, among all revised labels candidates that are consistent, the one with a large inter-class distance and a small intra-class distance is preferred.

Calculating the similarity between instances does not depend on any supervised training. Hence, one could use any unsupervised learning approaches to obtain a good $\phi$ without accessing any label information [3, 7]. This consistency measure can be regarded as an implicit clustering initialization on the output space of function $\phi$ [29]. In addition, if we use the perception model $f$'s embedding layer as the function $\phi$, the $\phi$ and the distance measurement could be further improved during the subsequent neuro-symbolic learning and accelerates the abduction reasoning.

## 5 Abductive Learning with Similarity

Based on the proposed consistency measure, we present the Abductive Learning with Similarity (ABLSim) approach, which is optimized through a greedy beam search. And then we show that ABLSim can be simply combined with other consistency measures to enhance abduction-based neuro-symbolic learning. Algorithm 1 shows an outline of ABLSim.

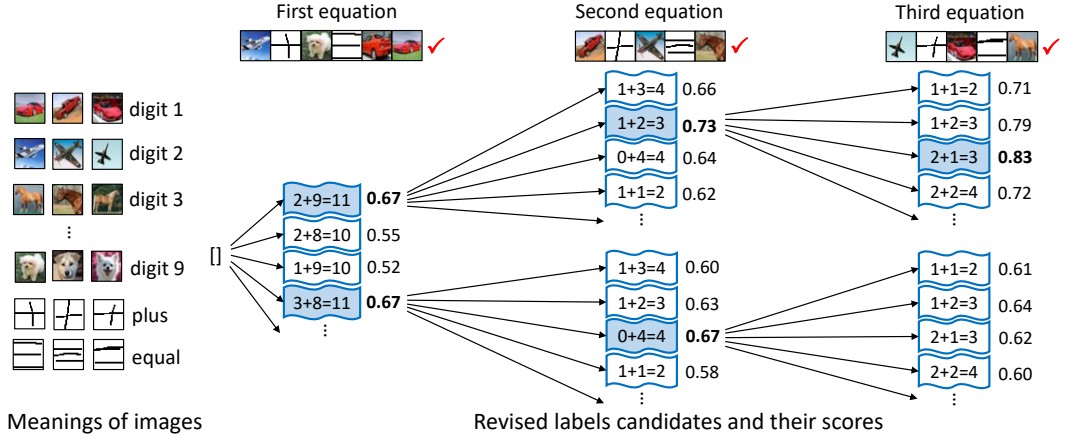

Figure 2: An example of beam search in ABLSim. It involves three unlabeled and consistent equations whose ground-truth labels are "2+9=11", "1+2=3", "2+1=3", and the beam width is 2. In the first iteration, after computing the scores, '2+9=11' and '3+8=11' are kept in the candidate set. In the second iteration, "2+9=11 1+2=3" and "3+8=11 0+4=4" rank top 2 among all candidate revised labels, while other candidates are pruned. In the last iteration, we get the final abduced label "2+9=11 1+2=3 2+1=3" according to the computed scores of consistency measure.

## 5.1 Optimization

One possible situation when applying ABLSim is that the number of instances in $x$ is usually small, and for some example $x^{\langle k \rangle}$ there is no $\langle i, j \rangle$ such that $\bar{z}_i^{\langle k \rangle} = \bar{z}_j^{\langle k \rangle}$, making it challenging to calculate the intra-class distance. For example in Figure 1(b), the size $|x| = 6$, and for the revised pseudo-labels candidate $\bar{z} =$"2+9=11", we could not measure the intra-class distance of the first image $x_1$ because there is no other sample $x_j$ whose revised label $\bar{z}_j$ is digit 2. Moreover, even if instances with the same revised label exist, the average distance may deviate due to limited samples.

To get a sound and reliable score for consistency measure, we borrow some more samples to conduct the abductive reasoning. Assume that there are $m$ examples, denoted as $x^{\langle 1 \rangle}, x^{\langle 2 \rangle}, \cdots, x^{\langle m \rangle}$. Each example $x^{\langle k \rangle}$ has a corresponding final output $y^{\langle k \rangle}$ and the candidate set of revised labels is $\mathbb{A}^{\langle k \rangle}$. If labeled data exists, it can also be regarded as bags of instances, whose candidate pseudo-label set $\mathbb{A}^{\langle k \rangle}$ contains only the ground-truth labels. Let $X = (x^{\langle 1 \rangle}, x^{\langle 2 \rangle}, \cdots, x^{\langle m \rangle})$ be the set of input examples, and $\bar{Z} = (\bar{z}^{\langle 1 \rangle}, \bar{z}^{\langle 2 \rangle}, \cdots, \bar{z}^{\langle m \rangle})$ be the set of corresponding bags of abduced labels. If $\bar{Z}$ is consistent with knowledge base, it should be an element of the Cartesian product of $\mathbb{A}^{\langle k \rangle}$, i.e., $\bar{Z} \in \mathbb{A}^{\langle 1 \rangle} \times \mathbb{A}^{\langle 2 \rangle} \times \cdots \times \mathbb{A}^{\langle m \rangle}$. The abduction problem for set $X$ can be formalized as follows:

$$\max_{\bar{Z}} \quad \text{Score}(X, \bar{Z}), \tag{7}$$

$$s.t. \quad X = (x^{\langle 1 \rangle}, x^{\langle 2 \rangle}, \cdots, x^{\langle m \rangle}), \tag{8}$$

$$\bar{Z} \in \mathbb{A}^{\langle 1 \rangle} \times \mathbb{A}^{\langle 2 \rangle} \times \cdots \times \mathbb{A}^{\langle m \rangle}, \tag{9}$$

$$\mathbb{A}^{\langle k \rangle} = \{\bar{z} \mid KB \cup \bar{z} \models y^{\langle k \rangle}\}. \tag{10}$$

This is a combinatorial optimization problem where the search space of $\bar{Z}$ grows exponentially with $m$. ABLSim uses beam search to solve this optimization problem greedily. Eventually, the element with the highest score in the final candidate set $\mathbb{A}$ will be selected as the best abduced labels.

Figure 2 gives an example of the beam search in ABLSim in the *CIFAR-10 Equation Decipherment* task. There are three unlabeled examples $x^{\langle 1 \rangle}, x^{\langle 2 \rangle}, x^{\langle 3 \rangle}$ consistent with the knowledge base and the beam width $b = 2$. In each iteration, ABLSim keeps the top-2 candidates ranked by their scores. The final abduced label is $\bar{Z} =$"2+9=11 1+2=3 2+1=3", which is the ground-truth label.

The above ABLSim algorithm could be accelerated by GPU and parallel computations. The basic idea is that the score of each candidate label $\bar{z}$ in $\mathbb{A}$ (cf. Line 9-10 in Algorithm 1) can be computed separately, and we formulate the calculation as matrix operations for parallel computations with GPU.

Table 1: Image test accuracy and convergence time of different methods. The convergence time stands for the time taken to accuracy 95% in handwritten images and 85% for CIFAR-10. NGS-dft stands for NGS with default parameters, and NGS-opt for the fine-tuned optimal parameters. We set the time limit to 10 hours for all tasks. ● indicates ABLSim is significantly better than compared methods (paired t-tests at 95% significance level)

| | Method | Addition | Addition (CIFAR) | HWF | HWF (CIFAR) |
|---|---|---|---|---|---|
| Acc / % | DeepProbLog | 96.5±0.5● | 21.6±1.7● | 32.2±0.6● | 15.2±2.6● |
| | NGS-dft | 39.9±54.1● | 38.7±35.1● | 99.6±0.2● | 23.8±6.3● |
| | NGS-opt | 98.5±0.3● | 88.7±0.8 | 99.6±0.2● | 66.0±14.5● |
| | ABLSim (ours) | **98.8±0.1** | **88.9±0.5** | **99.9±0.1** | **88.4±0.7** |
| Time / s | DeepProbLog | 396±3 | time out | time out | time out |
| | NGS-dft | time out | time out | 299±36 | time out |
| | NGS-opt | 46±4 | 6954±558 | 240±7 | time out |
| | ABLSim (ours) | **42±5** | **6066±79** | **130±4** | **7263±122** |

During the beam search, the score of each $\bar{z}$ is updated incrementally based on the previous results, which further reduces the memory and time consumption. Please refer to the supplementary material for details.

## 5.2 Combining Similarity and Confidence in Consistency Measure

ABLSim can also be combined with other consistency measures. For example, the confidence score of a revised pseudo-label $\bar{z}$ could be defined as follows:

$$\text{ConfidenceScore}(\boldsymbol{x}, \bar{\boldsymbol{z}}) = \frac{1}{|\boldsymbol{x}|} \prod_{x_i \in \boldsymbol{x}} \text{Confidence}(x_i, \bar{z}_i), \tag{11}$$

where $\text{Confidence}(x_i, \bar{z}_i)$ indicates the confidence that sample $x_i$ belongs to label $\bar{z}_i$.

Combining (2) and (11), we get the final score for ABLSim's consistency measure:

$$\text{Score}(\boldsymbol{x}, \bar{\boldsymbol{z}}) = \theta \cdot \text{SimilarityScore}(\boldsymbol{x}, \bar{\boldsymbol{z}}) + (1 - \theta) \cdot \text{ConfidenceScore}(\boldsymbol{x}, \bar{\boldsymbol{z}}), \tag{12}$$

where $\theta \in \mathbb{R}$ is the weighting coefficient. The similarity score and confidence score need to be normalized before computing the final score.

## 6 Experiments

This section presents the experimental results on four neuro-symbolic tasks, including two benchmark datasets and two hard tasks with increased perception difficulty, to demonstrate that ABLSim can perform more efficient and effective abduction than previous state-of-the-art methods by leveraging the similarity among samples. Furthermore, we verify whether it can help the perception model learn faster and achieve better performance with less labeled data and weaker domain knowledge. All experiments are repeated five times on a server with Intel Xeon Gold 6248R CPU and Nvidia Tesla V100S GPU. In experiments, we simply fix the coefficient $\theta$ of ABLSim to 0.96, though it may be better to let it vary with training. The hyperparameters of ABLSim are determined by cross-validation on training data. The code is available for download[1].

### 6.1 MNIST (CIFAR-10) Addition

This task was first introduced in [21], the inputs are pairs of MNIST [17] images and the outputs are their sums. We also prepare a hard version of this task by replacing the MNIST images with CIFAR-10 [16] images. We compare ABLSim with the Neural-Grammar-Symbolic (NGS) model [19] (with default and fine-tuned parameters) and DeepProbLog [21]. All methods share the same knowledge base and perception model (LeNet [17] for MNIST and ResNet-50 [12] for CIFAR-10), which are initialized randomly. The feature for calculating the similarity score is obtained from the second last layer in the perceptual neural net.

Table 1 shows the experimental results. In the MNIST Addition task, all methods except NGS-dft converge. DeepProbLog converges slowly because computing gradients of the probabilistic logic

---
[1]https://github.com/AbductiveLearning/ABLSim

program is time-consuming. The performance of NGS-dft is greatly affected by initialization, which leads to a large standard deviation in the table. For the CIFAR-10 version, DeepProbLog and NGS-dft fail to converge, while NGS-opt converges slower than ABLSim, which solves all tasks efficiently and achieves higher accuracy.

## 6.2 Handwritten Formula Recognition

The HWF dataset [19] contains images of decimal formulas and their computed results. We also create a (perceptual) harder version by replacing the digit images with CIFAR-10 [16] images.

The perception neural networks are initialized randomly for all methods. Note that the HWF in the original paper of NGS includes length-one equations as a pre-training curriculum, which are deleted in our experiments. Thus our results are different from those in [19].

As shown in Table 1, ABLSim significantly outperforms others by time efficiency and predictive accuracy. Because the background knowledge is the same, this result verifies the benefit of leveraging the similarity-based consistency optimization. Both NGS and DeepProbLog are trapped in local minimum and fail to converge in the hard version tasks, this is probably caused by the unreliable confidence output from the perception neural network when it is not pre-trained.

## 6.3 CIFAR-10 Decimal Equation Decipherment

### 6.3.1 Dataset and Knowledge base

This task is the most challenging one in our experiments, which extends the Digital Binary Additive equations experiment in ABL [5] with *decimal* digits. The input contains images of an arithmetic equation and the output is the label of its correctness. Instead of learning binary equations represented by MNIST images, here we use CIFAR-10 [16] images to encode decimal equations. An example of this dataset is shown in Figure 2. The task's inputs are sequential images of randomly generated decimal equations, which consist of twelve symbols $(0,1,\cdots,9,+,=)$. Besides, the knowledge base contains symbolic rules about how to carry out decimal addition operations. The correctness labels ($True$ or $False$) of training equations are provided in the form of logical facts.

### 6.3.2 Experimental Setup

We compare ABLSim with four baselines: 1) **ABL**: Abductive learning [5] that maximizes consistency with minimal revisions. If multiple abduced labels have the minimal revisions, it would randomly select one. 2) **ABL-Conf**: Abductive learning that measures consistency using only the confidence predicted by neural net, i.e., $\theta = 0$ in equation (12). 3) **ABL-pre-train**: ABL with model pre-trained on a small set of labeled images. 4) **ABL-Conf-pre-train**: ABL-Conf with a pre-trained network. ABLSim uses the perception model's embedding layer to calculate the similarity. For the non-pre-trained methods, including ABLSim, the perception model, a ResNet-50 [12], is initialized by self-supervised learning [3] on training images. Note that this process is unsupervised and does NOT use any image labels.

### 6.3.3 Results

**Learning Curve.** The learning curves of all models are shown in Figure 3(a). The proposed ABLSim converges much faster and achieves higher accuracy than other methods. ABL-Conf and ABL fail to converge and get low accuracy because their abduction processes are based on the pseudo-labels or label confidence predicted by the perception model, which are unreliable when there lacks supervised pre-train. Both ABL and ABL-pre-train are inferior to their '-Conf' versions, indicating that pseudo-label confidence is helpful. Starting from high initial accuracy, ABL-Conf-pre-train gets a similar accuracy as ABLSim, but it converges slower than our presented ABLSim, which learns without any supervised pre-training and has the fastest converge rate.

**Rate of Successful Abduction.** Figure 3(b) illustrates the curves of different methods' abduction success rate, i.e., the proportion of the correctly abduced labels. ABLSim converges faster and achieves a nearly optimal result, where the abduced labels are exactly the ground-truth labels. In the first few iterations, the abduction success rate is relatively low. However, the extracted embeddings of

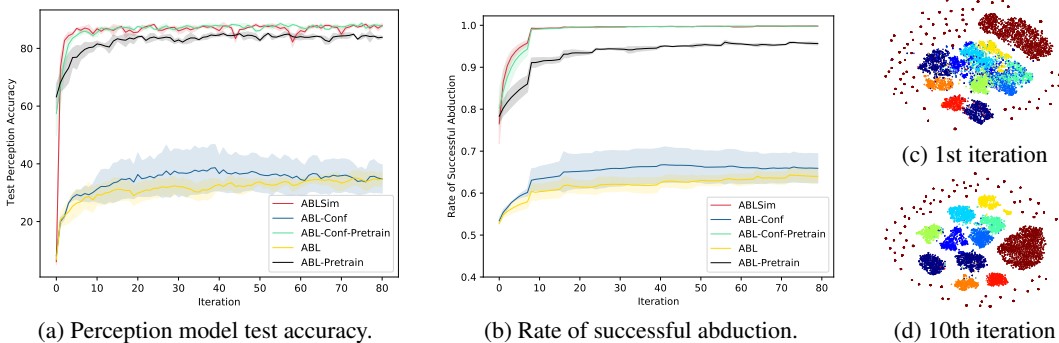

| (a) Perception model test accuracy. | (b) Rate of successful abduction. | (c) 1st iteration |
| | | (d) 10th iteration |

Figure 3: Learning curves (a & b) and the t-SNE visualization of the learned embeddings (c & d).

Table 2: The rate of successful abduction w.r.t. different beam width in the first iteration.

| Beam Width | 16 | 32 | 64 | 128 | 256 | 512 | 1024 | 2048 |
|---|---|---|---|---|---|---|---|---|
| Successful Abductions | 62% | 63% | 65% | 71% | 72% | 73% | 75% | 75% |

images are gradually improved during training, which provide a better similarity metric for abductive reasoning. Therefore, the rate of successful abduction reaches 1.0 just in a few iterations.

**Embedding Visualization.** To verify whether the extracted embeddings of neural network are improved during training, we plot t-SNE [26] visualizations on the second last layer of neural network that we use for calculating the consistency score. As shown in Figure 3(c)-(d), in the beginning, the embeddings of classes are not well separated. However, during abductive learning, the distance between the different classes becomes larger after the neural net is updated with the abduced labels, which will in turn help accelerate the abduction.

**Influence of Beam Width.** We study the influence of beam width in optimization, and the results are presented in Table 2. The successful abduction increases as beam width becomes larger, while the time cost grows linear at the same time. The improvement becomes marginal when the beam width is larger than 1k. We use a beam width of 600 in our implementation to achieve the balance between convergence rate and time complexity.

## 6.4 Theft Judicial Sentencing

Table 3: Micro-F1-score of the model, and MAE of the predicted sentence. The label rates are denoted as suffixes.

| KB | Method | F1 | MAE |
|---|---|---|---|
| N/A | PL-10 | 0.814 | 0.862 |
| N/A | Tri-10 | 0.812 | 0.840 |
| Full | SS-ABL-10 | 0.862 | 0.824 |
| Full | ABLSim-10 | 0.861 | 0.825 |
| Part | SS-ABL-10 | 0.833 | 0.835 |
| Part | ABLSim-10 | **0.851** | **0.828** |
| N/A | PL-50 | 0.858 | 0.832 |
| N/A | Tri-50 | 0.861 | 0.810 |
| Full | SS-ABL-50 | 0.865 | 0.788 |
| Full | ABLSim-50 | 0.866 | 0.786 |
| Part | SS-ABL-50 | 0.862 | 0.803 |
| Part | ABLSim-50 | **0.866** | **0.783** |

This is a real-world task in [13], which aims to train a model that outputs sentencing elements from criminal judgment texts, and optimize the parameters in knowledge base to predict the sentence of a defendant. We use the same knowledge base and BERT model [7] as in [13], where the knowledge base contains many law articles represented by logic rules. We additionally make the task more challenging by removing some pre-defined rules to weaken the background knowledge base [13].

We compare ABLSim with SS-ABL [13] and two semi-supervised methods Pseudo-Label (PL) [18] and Tri-training (Tri) [30]. It takes about 1 hour to train the model. As shown in table 3, ABL-based models are superior to the semi-supervised methods. When the number of labeled pre-train data is small (10%), the performance of ABLSim with weaker KB is even comparable to the SS-ABL with a full KB. When the number of labeled pre-train data increased to 50%, ABLSim achieves the highest performance with less pre-defined domain knowledge.

# 7   Conclusion

In this paper, we propose a novel consistency measure for abduction-based neuro-symbolic learning, and further present the Abductive Learning with Similarity (ABLSim) method. Compared to previous approaches, it not only exploits the full-featured logical reasoning ability, but also utilizes the information hidden in the feature space of the input data. The proposed consistency measure leverages the similarity between samples, which previous approaches have never considered. Empirical evaluation validates that ABLSim significantly outperforms the state-of-the-art neuro-symbolic learning approaches in terms of speed and performance with less labeled data. The proposed consistency measure is general, and any other abduction-based approach can use it. The limitation of the method is that the rules of the domain knowledge base need to be constructed manually. In future work, we will try to incorporate techniques like predicate invention into our approach to discover new classes and new knowledge, so that we can automatically extend the knowledge base.

## Acknowledgments and Disclosure of Funding

This work is supported by the National Key Research and Development Program of China No.2020AAA0109400. Wang-Zhou Dai acknowledges the financial support from the UK's EP-SRC Robot Synthetic Biologist Project (EP/R034915/1). Stephen H. Muggleton wants to thank the support from EPSRC Network on Human-Like Computing (EP/R022291/1).

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
