# A  Appendix

## A.1  Details of calculating the Similarity Score

We give an example of the calculating the $\mathrm{IntraclassDis}$ in $\mathrm{SimilarityScore}$ when the denominator becomes $0$. Suppose the intra-class distance of each $x_i$ is $[0.1, 0.2, nan, 0.6]$, where $nan$ means the denominator is $0$, then we replace $nan$ with the average intra-class distance $(0.1+0.2+0.6)/3 = 0.3$ and the distance becomes $[0.1, 0.2, 0.3, 0.6]$.

## A.2  Acceleration of Consistency Score Calculation

As described in the paper, our proposed ABLSim algorithm could be accelerated by GPU and parallel computations. In our implementation, we use the GPU version of CuPy[1] to calculate the score.

We also develop an incremental calculation algorithm that calculates the scores based on previous calculation results in beam search. To calculate the score of consistency measure, ABLSim needs to compute the average inter-class distance and intra-class distance with respect to the revised label $z$. Assume that the label $z^{pre}$ is part of $z$, and we have computed its score in the previous iteration in beam search. ABLSim records the sum and the number of instances when calculating the previous average distance, and updates them incrementally in the next iteration. This operation dramatically reduces the need for memory space and lowers the time complexity.

## A.3  Complexity

The time complexity of beam search algorithm is $O(m * b * setsize)$ ($m$: input size; $b$: beam width; $setsize$: the size of the candidate set of revised labels), and the space complexity is $O(b * setsize)$, which are both polynomial.

The complexity for abducing candidates could be exponential in the worst case. However, the search space could be pruned by abduction and leveraging the similarity and confidence score. The implementation of ABLSim could be further accelerated conveniently by engineering techniques like parallel computing and caching. For example, in the experiment, we store the abduced results (valid solutions) in the cache to speed up the already fast constrained problem solving. This is feasible because the domain knowledge in our experiments is fixed.

The exponential time complexity is an inevitable problem in neuro-symbolic (NeSy) learning, which assumes no direct supervision for training the perceptual model and only has labels of the final output. The pseudo-label search is intrinsically NP-hard. To make the training possible, end-to-end NeSy models usually require exponential numbers of labeled samples; while hybrid neural logical models use domain knowledge and perform optimization in the exponential hypothesis space. They usually navigate in this space just by the probabilistic distribution output from the perceptual neural net, which could be unreliable when the neural net is under-trained. That's why the exponential complexity problem slows down their learning process.

## A.4  Experimental Details

### A.4.1  Decidability of Knowledge Base

The knowledge base used in our experiments is described by definite clauses, and the domain is finite since the number of abducible groundings is finite. Hence the logical abduction of ABLSim in our experiments is decidable.

### A.4.2  Neural Network & Hyperparameters

All compared methods share the same neural network structure for the same dataset. The neural networks in our experiments are implemented by PyTorch[2]. The BERT [2] model is implemented by TensorFlow[3].

---

[1] https://cupy.dev/
[2] https://pytorch.org/
[3] https://www.tensorflow.org/

- **MNIST (CIFAR-10) Addition**. For the MNIST addition task, we use the same neural network structure and the same hyper-parameters as in DeepProbLog[4] [7]. For the CIFAR-10 addition task, we use the ResNet-50 [3] structure as SimCLR[5] [1]. The training data of CIFAR-10 are augmented by applying random cropping (followed by resizing back to the original size), random color distortions, and random flip. We use the Adam [5] optimizer and its default hyperparameters. In abduction, the features used in ABLSim are the outputs from the penultimate layer of neural networks. We set the weighting coefficient $\theta = 0.96$ and beam width $b = 600$.

- **Handwritten Formula Recognition**. For the handwritten images, we use the same neural network structure as in the open-source code[6] of NGS [6]. The default hyperparameters have low performance in the experiments. We fine-tune the hyperparameters and use a learning rate of $10^{-3}$ and a search step of $50$. For the CIFAR-10 version, we use the same ResNet-50 [3] structure and hyperparameters as in our previous experiments.

- **CIFAR-10 Decimal Equation Decipherment**. The dataset contains 16,384 training equations and 4,096 testing equations in total. We use the same ResNet-50 [3] structure and hyperparameters as in our previous experiments, while the self-supervised model weights are trained by SimCLR[5] [1].

- **Theft Judicial Sentencing**. The dataset contains 687 court records of theft happened in 2017—2018, which has been anonymized with no possible way to identify a person. There are eight kinds of sentencing element labels, including *recidivism*, *confession*, *surrender*, *juvenile*, *forgiveness*, *no loss*, *pickpocket*, *burglary*. Our implementation is based on the source code of SS-ABL[7] [4] and we use the same knowledge base. We employ the multi-label Bidirectional Encoder Representations from Transformers (BERT) [2] as the classifier, a state-of-the-art model for Natural Language Processing (NLP). We use the pre-trained model and default hyperparameters in the official implementation[8].

### A.4.3 Learning Curves

We show the learning curves of ABLSim in the MNIST (CIFAR-10) Addition and Handwritten Formula Recognition experiments. As shown in Figure 1 and Figure 2, the curves on the left represent handwritten images experiments, while the curves on the right represent CIFAR-10 images experiments. The blue line is the abduction success rate (the proportion of the correctly abduced labels); the red line is the test accuracy of the perceptual model. Shaded regions indicate standard deviation.

As we can observe, since the task of handwritten images is simple, both model test accuracy and abduction success rate increase fast and reach nearly optimal. For the CIFAR-10 tasks, the abduction success rate is greater than the model test accuracy, because it is hard for the perception model to learn from CIFAR-10 images. Nevertheless, the model reaches a high test accuracy at the end.