# OpenReview forum: "Fast Abductive Learning by Similarity-based Consistency Optimization"
_NeurIPS.cc/2021/Conference — NeurIPS 2021 Poster_

### Official Review · Reviewer_Nytp · 2021-07-11

**Rating:** 4
**Confidence:** 4

**Summary:**

In the setting of neuro-symbolic approaches, this paper defines new consistency measures for improving the performance of abductive learning. The overall ABLSim framework uses two key components: a perception model $f$ that maps raw data instances $\mathbf x$ into symbolic instances $\mathbf z$, and a knowledge base $KB$ used to reason about symbolic instances $\vec z$, together with observed labels $y$. The goal of abductive learning is to revise the predictions made by the perception model $f$ according to the logic reasoning; notably, if a given label $y$ cannot be a logical consequence of $KB$ and a predicted instance $\mathbf z$, then by abduction the knowledge base $KB$ provides a set of new candidates $\mathbb A$. Since $\mathbb A$ can include multiple candidates, the key idea of this study is to choose the best ones using a ranking approach (Algorithm 1), together with consistency measures combining similarity scores and confidence scores (12). The ABLSim framework is validated in various experiments.


**Ethical Concerns:**

It seems that the "Theft Judicial Sentencing" task, examined in Section 6.4. and further detailed in the SM can raise some ethical issues. Indeed, the dataset contains 687 court records of theft, which happened in China during 2017–2018. I think that this dataset has been anonymized with no possible way to identify a person, but this should be explicitly written in a revised version of the paper.  Since the authors have used the knowledge base in [6,12], I believe that this base does not include any reference to some person. But again, this ethical point should be explicitly written in the revised version.

**Limitations And Societal Impact:**

As mentioned by the authors, the fact that $KB$ must be given in advance is a limitation in abductive learning. Based on the preview,  I would also say that an exponential space algorithm is also a key limitation.

**Main Review:**

The topic of abductive learning is well-motivated, and the experimental results look promising. Unfortunately, I am not convinced that the paper can be accepted in its current state, due to many technical issues.

(i) The key bottleneck of abductive reasoning lies in the complexity of inferring and enumerating explanations. Virtually nothing is said about these critical issues. Namely, given a first-order knowledge base $KB$ and a fact $y = 1$, what is the computational complexity of inferring a candidate $\mathbf z$ such that $KB \land \mathbf z \models (y = 1)$? To this very point, I guess that the authors have used a restricted (and decidable) fragment of first-order logic for the language upon which $KB$ is defined. More generally, since $\mathbb A$ can be of exponential size (this is indeed the case even for propositional Horn formulas), does there exist an output polynomial algorithm for enumerating from $KB$ the candidates in $\mathbb A$?

(ii) Several notions are incorrectly defined or left unspecified. For example, what is the beam width $b$ in Algorithm 1? What is the meaning of the expression $\mathrm{TopN}(\mathbb A, \mathit{score}, b)$? What is the meaning of $\mathrm{Update}(f, \mathbf X, \mathbf Z)$ ... Namely, how do we update the learning model $f$? When the denominator in (3) or (4) is zero, the expression (5) is left undefined. The explanation “we use the average inter-class/intra-class distance of other $x_j$ as the estimated distance of $x_i$” is a bit confusing; it would be preferable to reformulate (5) using the equality in Line 156.


(iii) Finally, Algorithm 1 requires exponential space, due to the fact that Line (5) computes sets $\mathbb A_k$ of (possibly) exponential size, together with the fact that Line (6) builds a Cartesian product of these sets. To this very point, I do not think that $\mathbb A_k$ should be stored, since each candidate can be evaluated on the fly according to (12). In other words, the exponential space required by Algorithm 1 is not necessary. This observation is connected to the idea of "parallelizing" the algorithm, which is suggested in the SM. But the algorithm should be rewritten properly in order to clarify the fact that its space complexity remains polynomial in the size of the input dataset and knowledge base.



**Time Spent Reviewing:**

3

---

> ### Author Response · Authors · 2021-08-06
> **Reply to Reviewer Nytp**
>
> Thanks for your review. We address your main concerns as follows.
>
> **[Q1]** The key bottleneck of abductive reasoning lies in the complexity of inferring and enumerating explanations (abduction).
>
> **[A1]** Thank you for pointing out this problem, which is common in almost all neuro-symbolic learning frameworks. We want to clarify that ABLSim **does not make simple enumeration that is inefficient**, and **it is a practical way to address this crucial problem**:
>
> 1. The exponential time complexity is an **inevitable problem** in neuro-symbolic (NeSy) learning, which assumes *no direct supervision for training the perceptual model* and only has labels of the final output. To make the training possible, end-to-end NeSy models usually require exponential numbers of labeled samples; while hybrid neural logical models (e.g., DeepProblog and NGS) use domain knowledge and perform optimization in the exponential hypothesis space. They usually navigate in this space just by the probabilistic distribution output from the perceptual neural net, which could be unreliable when the neural net is under-trained. That's why the exponential complexity problem slows down their learning process.
> 2. ABLSim uses logical abduction to **prune the search space** (based on constrained problem solving, which is much faster than simply enumerating all possible abductions). Furthermore, by leveraging the similarity and confidence score, ABLSim can further accelerate the abduction. Therefore, when using the **identical domain knowledge** in the experiments, our method achieves the highest time efficiency in comparison with the state-of-the-art method.
> 3. Logical abduction is **faster** than other compared methods (e.g. random walk based solely on a potentially unreliable probability distribution). ABLSim's logical abduction is actually implemented by constrained problem solving aided by domain knowledge, together with the training labels of the final output, they can navigate the optimization process pretty fast. This is validated by our experimental results.
> 4. The implementation of logical abduction **can be further accelerated conveniently** with simple engineering tweaking such as caching and parallel computing. For example, in the experiment, we store the abduced results (valid solutions) in the cache to speed up the already fast constrained problem solving. Parallel computing with GPU is also possible, please refer to our [reply A3](https://openreview.net/forum?id=UMrf6F4Tg9c&noteId=0VqRd7ZtD5) to reviewer jmHQ.
> 5. As we have mentioned before, the exponential search space is most likely inevitable for neuro-symbolic learning. One could use massive labeled training data and a well-defined environment for reinforcement learning to perform this search implicitly; or we could utilize background knowledge to prune the search space explicitly by abduction. Some promising directions to address this problem fundamentally include proving P=NP or using quantum computing.
>
> **[Q2]** Several notions are incorrectly defined or left unspecified... The explanation “we use the average inter-class/intra-class distance ..." is a bit confusing. It would be preferable to reformulate (5)...
>
> **[A2]** Thanks for your detailed comments. We will improve the presentation and make the terminology more clear.
>
> - The beam width $b$ is the beam size of beam search.
> - The meanings of the two expressions have been briefly illustrated in the annotations in Algorithm 1 (cf. Line 12,15 in Algorithm 1).
>
>   - TopN$(\mathbb{A}, \boldsymbol{score}, b)$ means selecting '$b$' elements in $\mathbb{A}$ with the highest $\boldsymbol{score}$.
>   - Update$(f, \boldsymbol{X}, \boldsymbol{\bar{Z}})$ stands for updating current model $f$ using data $\boldsymbol{X}$ and abduced labels $\boldsymbol{\bar{Z}}$, e.g. backpropagation algorithm for neural networks.
>
> We give an example of the explanation in Line 157-158. Suppose the intra-class distance of each $ x_i $ is $ [0.1, 0.2, nan, 0.6] $, where $ nan $ means the denominator is zero, then we replace $ nan $ with the average intra-class distance $ (0.1+0.2+0.6)/3=0.3 $ and the distance becomes $ [0.1, 0.2, 0.3, 0.6] $.
>
> **[Q3]** Algorithm 1 requires exponential space ... I do not think that $ \mathbb{A}^{\langle k\rangle} $ should be stored, since each candidate can be evaluated on the fly.
>
> **[A3]** Thank you for pointing this issue out. This is in fact a time-memory trade-off. Storing $ \mathbb{A}^{\langle k\rangle} $ makes the following parallel computations feasible and speeds up the algorithm. Besides, the space can be largely pruned as stated in A1.
>
> **[Q4]** I think that this dataset has been anonymized, but this should be explicitly written in a revised version of the paper.
>
> **[A4]** Thank you for the constructive comments. Yes, the dataset has been anonymized with no possible way to identify a person. We will clarify this in the revised version.

---

> > ### Comment · Reviewer_Nytp · 2021-08-29
> > **Re: Reply**
> >
> > Thanks for your detailed response.
> >
> > So, based on your response, I think that the clarity issue related to notations/definitions will be fixed.
> >
> > Yet, I am still concerned by the decidability and complexity issues.  For the moment, we do not even know if the knowledge base is described in a decidable fragment of first-order logic. Furthermore, the space/time complexity of ABLSim is not clear. On the one hand, the algorithm requires exponential space (and hence exponential time) due to the fact that it stores a Cartesian product of enumeration sets. On the other hand, as indicated by a fellow reviewer, ABLSim is using beam search. So, if it using only restricted samples of enumeration sets, its runtime complexity could be controlled. To sum up, the technical part of the paper should be revised in order to have a clear, unambiguous idea of the decidability/complexity aspects of the framework.

---

> > > ### Author Response · Authors · 2021-08-30
> > > **Follow-up Answers**
> > >
> > > **[Q5]** About decidability.
> > >
> > > **[A5]** Thank you for pointing out this problem, and we will make it clear in revision. The knowledge base used in our experiments is described by definite clauses, and the domain is finite since the number of abducible groundings is finite. Hence the logical abduction of ABLSim in our experiments is decidable.
> > >
> > > **[Q1+]** About complexity.
> > >
> > > **[A1+]**
> > >
> > > This is indeed an important problem requiring further clarification, and a discussion on complexity will be added to the revised paper.
> > >
> > > 1. The time complexity of beam search algorithm is $O(m * b * setsize)$ ($m$: input size; $b$: beam width; $setsize$: the size of the candidate set of revised labels), and the space complexity is $O(b * setsize)$, which are **both polynomial**.
> > > 2. The complexity for abducing candidates could be exponential in the worst case. However, the search space **could be pruned by** abduction and leveraging the similarity and confidence score. The implementation of ABLSim **could be further accelerated conveniently** by engineering techniques like parallel computing and caching. The caching trick used in our implementation is a widely-used acceleration method in the first-order logic inference of classic AI, which is named as "*tabling*" in any Prolog implementation.
> > > 3. As we have clarified before, the exponential complexity problem is **a problem for all neuro-symbolic learning systems** because the pseudo-label search is **intrinsically NP-hard**. We think it is unfair to criticize ABLSim for this reason, because it offers a practical way to solve this problem. Different to the embedding-based end2end models that require massive labeled examples, ABLSim uses **full-featured first-order logic** inference to abduce the pseudo-labels, which **guarantees the soundness and completeness**, and our experiments have verified that ABLSim is more efficient than the other current neuro-symbolic methods.

---

### Official Review · Reviewer_jmHQ · 2021-07-14

**Rating:** 6
**Confidence:** 5

**Summary:**

This paper works on the topic of neural-symbolic learning by abductive reasoning. Specifically, the task is to learn a perception model, given symbolic background knowledge and weak supervision. The authors propose a novel abduction strategy to speed up the search in abduction, which is based on the similarity between samples, rather than the output information from the perceptual model used in previous methods. Based on this strategy, this paper presents a neural-symbolic learning framework called "ABductive Learning with Similarity (ABLSim)", which uses beam search to approximate the combinatorial search problem in finding the abduced labels during training. The authors conduct extensive experiments on four tasks: MNIST Addition, Handwritten Formula Recognition, CIFAR-10 Decimal Equation Decipherment, and Theft Judicial Sentencing. The experimental results show that ABLSim is much more efficient than the state-of-the-art neural-symbolic learning methods.


**Limitations And Societal Impact:**

The comparison between ABLSim and baselines is a little unfair because ABLSim uses pretrained feature extractor and multiprocess parallel computations to speed up the search in abduction. The authors might perform a more fair comparison as suggested in the main review.

**Main Review:**

Strengths:
1. It is interesting and novel to develop a similarity-based consistency measure to guide the abduction process. The main idea is that after the abduction, the samples in the same (pseudo-)category should be similar in the feature space while samples from different categories should be not.

2. The derivation of the similarity-based consistency measure in Section 4 is reasonable and sound. Here unsupervised or self-supervised learning approaches can be used to obtain a good feature extractor without using any labels.

3. As shown in Table 1, the proposed framework ABLSim outperforms the state-of-the-art methods in terms of both accuracy and efficiency.

4. The paper is overall well-written and easy to follow.

Weaknesses:
1. My major concern is that the comparison between ABLSim and baselines (DeepProbLog, NGS) is a little unfair because ABLSim uses an extra pretrained feature extractor based on contrastive learning (SimCLR). This feature extractor provides an implicit clustering initialization and makes the learning easier in ABLSim, especially for CIFAR-10. I suggest that a more fair comparison be to also pretrain the perceptual model in baselines by unsupervised learning approach [1,2,3], such as SCAN [1], which is also based on SimCLR. FYI, SCAN can obtain ~88% accuracy on CIFAR-10 without using any labels during training.

[1] Van Gansbeke, Wouter, Simon Vandenhende, Stamatios Georgoulis, Marc Proesmans, and Luc Van Gool. 2020. “SCAN: Learning to Classify Images without Labels.” In ECCV. http://arxiv.org/abs/2005.12320.

[2] Ji, Xu, João F. Henriques, and Andrea Vedaldi. 2019. “Invariant Information Clustering for Unsupervised Image Classification and Segmentation.” In ICCV. http://arxiv.org/abs/1807.06653.

[3] Park, Sungwon, Sungwon Han, Sundong Kim, Danu Kim, Sungkyu Park, Seunghoon Hong, and Meeyoung Cha. 2020. “Improving Unsupervised Image Clustering With Robust Learning.” arXiv [cs.CV]. arXiv. http://arxiv.org/abs/2012.11150.

2. The computation efficiency of ABLSim is a little surprising to me (if I don't miss important details). As defined by Line 147, isn't $A$ exponentially huge in terms of the length of $z$? If so, shouldn't the max operation in Eq.(1) be very time-consuming? The Cartesian product in Line 6 of Algorithm 1 would make it further slower. The authors state in Line 203~205 that ABLSim could be accelerated by GPU and parallel computations. Could the authors elaborate on it? Besides, how much efficiency gain is brought by the parallel computations during beam search? I quickly went through the submitted code and found that multiprocess parallel is used during abduction ("abducer.py"). I was wondering how ABLSim would perform without such multiprocess parallel computations.

**Time Spent Reviewing:**

10

---

> ### Author Response · Authors · 2021-08-06
> **Reply to Reviewer jmHQ**
>
> Thanks for the detailed review and helpful comments. We answer your main questions as follows.
>
> **[Q1]** The comparison is a little unfair because ABLSim uses an extra pretrained feature extractor based on contrastive learning.
>
> **[A1]** We would like to clarify that the comparison is fair, because ABLSim does **not** use contrastive learning to initialize the neural network (cf. Line 227-228) in the experiment of both MNIST (CIFAR-10) Addition and Handwritten Formula Recognition (sorry for the unclear statement of initialization in the second experiment). The second last layer of neural network classifier serves as the feature extractor. We only use contrastive learning in the third (Decimal Equation Decipherment) experiment. Thanks for your suggestions, and it would be interesting to use the three mentioned unsupervised learning approaches to pretrain the perceptual model. We will add to discussion and references in the revision.
>
> **[Q2]** Isn't $\mathbb{A}$ exponentially huge in terms of the length of $\boldsymbol{\bar{z}}$? If so, shouldn't the max operation in Eq.(1) and the Cartesian product in Algorithm 1 be very time-consuming?
>
> **[A2]** This is an inevitable problem for all neuro-symbolic methods, either for end2end or for hybrid models. Abduction can make use of domain knowledge to perform a "back-propagation-like" constraint problem solving to prune the exponential search space for $\boldsymbol{\bar{z}}$. In practice, the size of $\mathbb{A}$ can be further smaller by restricting $\boldsymbol{\bar{z}}$ not too far away from the machine learning model's perceived results. In all our experiments, most of $\mathbb{A}$ has a size of less than 100. Besides, the max operation and Cartesian product can be accelerated by parallel instructions. Therefore, the max operation and the Cartesian product would not be time-consuming (occupies <2% time cost within an iteration). For more details, please refer to our [reply A1](https://openreview.net/forum?id=UMrf6F4Tg9c&noteId=0KNubvGhOx5) to reviewer Nytp.
>
> **[Q3]** Could the authors elaborate on (GPU) parallel computations? How much efficiency gain is brought?
>
> **[A3]** Some details of parallel computations have been included in the supplementary material. The basic idea is that the score of each candidate label $\boldsymbol{\bar{z}}$ in $\mathbb{A}$ (cf. Line 9-10 in Algorithm 1) can be computed separately, and we formulate the calculation as matrix operations for GPU parallel computations ("similarity\_calculator.py"). The efficiency gained by the parallel computations is similar to that of the training of neural networks by GPU, since both of them mainly consist of matrix operations. We will add more details according to your suggestions. Thank you.

---

> > ### Comment · Reviewer_jmHQ · 2021-08-27
> > **The CIFAR-10 Decimal Equation Decipherment experiment**
> >
> > Thanks for the responses.
> >
> > By checking Table 1, it seems that ABLSim is significantly better than NGS-opt **only on the HWF (CIFAR) task**. However, in this task, ABLSim uses contrastive learning to pretrain the feature extractor, while NGS-opt does not. It still looks unfair to me. What are the results of NGS-opt using contrastive learning on the HWF (CIFAR) task?

---

> > > ### Author Response · Authors · 2021-08-28
> > > **Follow-up Answer**
> > >
> > > **[Q1+]** On the HWF (CIFAR) task, the comparison is unfair because ABLSim uses contrastive learning to pretrain the feature extractor, while NGS-opt does not.
> > >
> > > **[A1+]** As we have clarified in **[A1]**, to make the comparison fair, ABLSim does **not** use contrastive learning to pre-train the neural networks in **all compared methods and tasks in Table 1**. All of them are randomly initialized. The feature for calculating the similarity score is obtained from the second last layer in the perceptual neural net.
> > >
> > > We **only use contrastive learning to pre-train ABLSim in the _Decimal Equation Decipherment experiment_** (Section 6.3) for ablation study, whose results are shown in **Figure 3**. It shows that a stronger pre-training can improve the learning efficiency in ABLSim by abducing more correct pseudo-labels.
> > >
> > >
> > > The results in Table 1 show that:
> > >
> > > 1. Without any pre-training, ABLSim is significantly better than NGS methods in those tasks (please refer to our [reply A1](https://openreview.net/forum?id=UMrf6F4Tg9c&noteId=rjfpr5faabW) to Reviewer ZiCC).
> > > 2. More importantly, a key contribution of ABLSim is it **converges significantly faster** than other compared methods.
> > >
> > >
> > > Thanks for pointing out this problem, we will make it clear in the revised paper.

---

### Official Review · Reviewer_QmdE · 2021-07-17

**Rating:** 7
**Confidence:** 4

**Summary:**

This paper proposes a new method for implementing abductive learning proposed by Wang-Zhou Dai et al. (NeurIPS 2019 [4]). The main difference lies in replacing the objective function that maximizes consistency with minimal revisions with a new objective function that maximizes a score for samples relevant to abduced labels, where the score considers both the confidence for samples being classified to abduced labels and the similarity between samples. Experimental results on four knowledge-enhanced datasets demonstrate that this revision on the objective function leads to the state-of-the-art performance and faster converge rate, whereas on the criminal judgement dataset [12] demonstrate that the proposed method achieves better performance with weaker domain knowledge.

**Limitations And Societal Impact:**

Yes. Almost all limitations are adequately addressed. A potential limitation may lie in the search space of abductive solutions. With complex domain knowledge, the search space of abductive solutions could be very large or even infinite. The current treatment that computes all abductive solutions before selecting some of them by beam search may not be practical for some scenarios.

**Main Review:**

Although the proposed method builds upon the established abductive learning framework [4], the training objective function is completely different and carefully verified by extensive experiments, thus the novelty of the paper is moderate. As far as I can see from the paper and the supplemented source code, the proposed method is technically sound and has sufficient experiments designed to justify the contribution parts. Since the abductive learning framework is a fundamental approach to integrating perceptual intelligence with cognitive intelligence, the development of a more efficient and effective method to implement the framework is very desirable and can have significant impacts. As for the clarity, the paper is well organized and well written except some minor questions remained to me. First, it is unclear how to determine the weighting coefficient between similarity score and confidence score. In particular, it is unclear how this coefficient is set in the experimental method ABLSim. Second, it is unclear whether the comparison method ABL-Conf is simplified from ABLSim by fixing the weighting coefficient to zero or is extended from ABL [4]. The paper says that ABL is inferior to its -Conf version, which means that ABL-Conf is extended from ABL, but it is unclear how to extend ABL to ABL-Conf. Finally, it is unknown how well ABLSim works on the criminal judgement dataset [12] with full domain knowledge. If it works worse with full domain knowledge than with partial domain knowledge, it should be very sensitive to domain knowledge selected. This would imply a limitation that the domain knowledge needs to be carefully introduced.

**Time Spent Reviewing:**

10

---

> ### Author Response · Authors · 2021-08-06
> **Reply to Reviewer QmdE**
>
> Thanks for your appreciation and constructive comments. We address your main questions as follows.
>
> **[Q1]** How to determine the weighting coefficient?
>
> **[A1]** In experiments, we simply fix the coefficient to 0.96, though it may be better to let it vary with training. The easiest way to determine this hyperparameter is using grid search and cross-validation.
>
> **[Q2]** Whether ABL-Conf is simplified from ABLSim by fixing the weighting coefficient to zero or is extended from ABL.
>
> **[A2]** ABL maximizes consistency with minimal revisions, and if multiple abduced labels have the minimal revisions, ABL would randomly select one. ABL-Conf selects the abduced labels with the maximal confidence score judged by the perceptual neural net. ABL-Conf can be extended from ABL by replacing random selection with confidence-based ranking; and it can also be simplified from ABLSim by fixing the weighting coefficient to zero. We will make it more clear in the revised version.
>
> **[Q3]** How well ABLSim works on the criminal judgement dataset with full domain knowledge?
>
> **[A3]** Thanks for your suggestion. We have made a quick experiment during rebuttal period, and found that there is no significant difference in performance between ABLSim between with full/partial domain knowledge base. The performance of previous methods (SS-ABL) drops after removing part of the knowledge, as shown in Table 3. This validates that less domain knowledge is needed for ABLSim in this task. We will clarify this in the revised version.
>
> **[Q4]** A potential limitation may lie in the search space of abductive solutions.
>
> **[A4]** Thanks for your question. Please see our [reply](https://openreview.net/forum?id=UMrf6F4Tg9c&noteId=0KNubvGhOx5) to the reviewer Nytp’s Q1.

---

> > ### Comment · Reviewer_QmdE · 2021-08-30
> > **Clarification on the complexity issue**
> >
> > I agree that the computational complexity of the proposed approach has been addessed to some degree in the submission, but I think the authors may mix up two different complexities according to their response to other reviewer's comments. The first complexity is for computing the complete set of abductive explanations (denoted $\mathbb A$ in the submission). This set $\mathbb A$ can be exponential in terms of the entities/constants considered in the domain, or even infinite when open domain is allowed. It is not  **a problem for all neuro-symbolic learning systems** but only for those ones storing and using the complete set of abductive explanations like the proposed method in the submission. I hope the authors will add some discussions in the revised paper to handle this issue, as suggested in my comment. The second complexity is for beam search on a sequence of complete sets of abductive explanations. It is clear that such a search is approximate and has a polymonial complexity. It is also noted that the computation of the complete set of abduction explanations can only guarantee the soundness and completeness before beam search. By beam search the true label is not guaranteed to be found in theory.

---

> > > ### Author Response · Authors · 2021-08-30
> > > **Follow-up Answers**
> > >
> > > **[Q4+]** About exponential search space.
> > >
> > > **[A4+]** Indeed, this kind of neuro-symbolic learning methods where pseudo-labels exist (such as ABL and DeepProbLog), face the problem of exponential complexity. However, ABLSim tries to prune the search through abduction and similarity, so it has a higher efficiency in the experiments.
> > >
> > > **[Q5]** About soundness&completeness.
> > >
> > > **[A5]** Beam search is indeed an approximate search strategy that finds local optimal. In real applications, we could use methods like random start to increase the probability of finding global optimal, which can also be accelerated in parallel.
> > >
> > > Thank you for your helpful comments. We will add some discussions to clarify these problems in the revision.

---

### Official Review · Reviewer_ZiCC · 2021-07-20

**Rating:** 7
**Confidence:** 4

**Summary:**

The paper proposes a similarity-based consistency measure for abduction, which takes the idea that samples in the same category are similar in feature space while samples of different classes are dissimilar.


**Limitations And Societal Impact:**

The rules in the BK should be manually defined

**Main Review:**

The authors propose ABductive Learning with Similarity (ABLSim), which adopts beam search to solve the optimization problem of finding the best abduced labels.

The paper is well written and the concepts are clear. The proposed approach is novel wrt the other neural symbolic approaches.

The experimental evaluation prove the validity of the proposed approach in term of time efficiency. As regards the accuracy score when compared to  Neural-Grammar-Symbolic model a test should be reported to assess the statistical difference.



**Time Spent Reviewing:**

3

---

> ### Author Response · Authors · 2021-08-06
> **Reply to Reviewer ZiCC**
>
> Thanks for your insightful comments. We address your concern as follows.
>
> **[Q1]** A test should be reported to assess the statistical difference.
>
> **[A1]** Thank you for your supportive comments and helpful suggestions. We will add the statistical difference in our revised version. Here we show the revised Table 1:
>
> Revised Table 1: Here we only show the accuracy part. ● indicates that ABLSim is significantly better than the compared methods (paired t-tests at 95% significance level).
>
> | Method        | Addition     | Addition (CIFAR) | HWF          | HWF (CIFAR)  |
> | :------------ | :----------- | :--------------- | :----------- | :----------- |
> | DeepProbLog   | 96.5±0.5 ●   | 21.6±1.7 ●       | 32.2±0.6 ●   | 15.2±2.6 ●   |
> | NGS-dft       | 39.9±54.1 ●  | 38.7±35.1 ●      | 99.6±0.2 ●   | 23.8±6.3 ●   |
> | NGS-opt       | 98.5±0.3 ●   | 88.7±0.8         | 99.6±0.2 ●   | 66.0±14.5 ●  |
> | ABLSim (ours) | **98.8±0.1** | **88.9±0.5**     | **99.9±0.1** | **88.4±0.7** |

---

### Decision · Program_Chairs · 2021-09-27

**Decision:**

Accept (Poster)

**Comment:**

There were two main concerns with this paper --- the algorithm requires exponential space and a concern about the fairness in the experimental results.  I am not worried about the worst case exponential space requirement of the exact algorithm.  Many algorithms have bad worst case behavior while having fast and effective approximations (as several reviewers note).  The one low score on this paper is due to the exponential space issue. Also, the discussion of the fairness of empirical comparisons convince me that this is an issue in the clarity of the writing rather than a fundamental issue with the paper.